# Functional hierarchies in brain dynamics characterized by signal reversibility in ferret cortex

Sebastian Idesis[1]*, Sebastián Geli[1], Joshua Faskowitz[2], Jakub Vohryzek[1,3], Yonatan Sanz Perl[1,4,5], Florian Pieper[6], Edgar Galindo-Leon[6], Andreas K. Engel[6], Gustavo Deco[1,7]

**1** Center for Brain and Cognition (CBC), Department of Information Technologies and Communications (DTIC), Pompeu Fabra University, Edifici Mercè Rodoreda, Barcelona, Catalonia, Spain, **2** Department of Psychological and Brain Sciences, Indiana University Bloomington, Bloomington, Indiana, United States of America, **3** Centre for Eudaimonia and Human Flourishing, Linacre College, University of Oxford, Oxford, United Kingdom, **4** National Scientific and Technical Research Council, Buenos Aires, Argentina, **5** Institut du Cerveau et de la Moelle épinière, ICM, Paris, France, **6** Department of Neurophysiology and Pathophysiology, University Medical Center Hamburg-Eppendorf, Hamburg, Germany, **7** Institució Catalana de Recerca I Estudis Avançats (ICREA), Barcelona, Catalonia, Spain

\* Sebastian.idesis@gmail.com

**Data Availability Statement:** Data and code are available at: https://github.com/SebastianIdesis/Hierarchy_Ferrets-2023-.

## Abstract

Brain signal irreversibility has been shown to be a promising approach to study neural dynamics. Nevertheless, the relation with cortical hierarchy and the influence of different electrophysiological features is not completely understood. In this study, we recorded local field potentials (LFPs) during spontaneous behavior, including awake and sleep periods, using custom micro-electrocorticographic (µECoG) arrays implanted in ferrets. In contrast to humans, ferrets remain less time in each state across the sleep-wake cycle. We deployed a diverse set of metrics in order to measure the levels of complexity of the different behavioral states. In particular, brain irreversibility, which is a signature of non-equilibrium dynamics, captured by the arrow of time of the signal, revealed the hierarchical organization of the ferret's cortex. We found different signatures of irreversibility and functional hierarchy of large-scale dynamics in three different brain states (active awake, quiet awake, and deep sleep), showing a lower level of irreversibility in the deep sleep stage, compared to the other. Irreversibility also allowed us to disentangle the influence of different cortical areas and frequency bands in this process, showing a predominance of the parietal cortex and the theta band. Furthermore, when inspecting the embedded dynamic through a Hidden Markov Model, the deep sleep stage was revealed to have a lower switching rate and lower entropy production. These results suggest functional hierarchies in organization that can be revealed through thermodynamic features and information theory metrics.

**Funding:** S.I is supported by the EU project euSNN (MSCA-ITN-ETN H2020-860563). G.D. received funding from the Horizon EU ERC Synergy Grant Project ID: 101071900; Spanish national research project (ref. PID2019-105772GB-I00/AEI/ 10.13039/501100011033) funded by the Spanish Ministry of Science, Innovation and Universities (MCIU); A.K.E. received funding from the DFG (SFB936-178316478-A2) and from the European Union (project cICMs, ERC-2022-AdG-101097402). The funders had no role in study design, data collection and analysis, decision to publish, or preparation of the manuscript. Views and opinions expressed in this paper are those of the authors only and do not necessarily reflect those of the European Union or the European Research Council. Neither the European Union nor the granting authority can be held responsible for them.

**Competing interests:** The authors have declared that no competing interests exist.

## Author summary

Understanding the brain functioning and the mechanisms underlying the transition between different brain states has been a key goal of modern clinical neuroscience. Global balance of the brain could be assessed by measuring spontaneous changes in brain dynamics. However, the subtlety of the effects demands advanced computational methods to extract the relevant dynamical information from neuroimaging recordings. To this aim, electrocorticographic measurements of ferrets provide a unique opportunity for inspecting these transitions in a long fluctuating recording. Our findings demonstrate the large, and still underexploited potential of several methods in the study of large-scale brain dynamics. We think that our approach can be fruitfully applied to a wide array of brain disorders, subserving both the theoretical goal of a clearer understanding of these diseases, and at the same time, the clinical goal of maximizing patients' classification, diagnosis, and prognosis. Finally, by providing detailed insight into the role different regions and states in global brain dynamics, our approach may inform external stimulation therapies that, combining with traditional behavioral therapies, may significantly accelerate recovery in different brain disorders.

## Introduction

The living brain is perpetually active, expending energy to support a wide array of behaviors, perception, and cognition. But can this ongoing brain activity be quantifiably studied, in a manner that aligns with observable behavioral states?

One way to answer this question is through the analysis of data acquired by electroencephalography (EEG) or functional magnetic resonance imaging (fMRI). By extracting time series features, the quantification of brain activity states, and their corresponding transitions, can be estimated. One promising analysis framework considers the description of brain states in terms of brain dynamics operating out of equilibrium, based on time series reversibility features [1–6]. Generally, in a non-equilibrium system, where the balance between the elements is broken, the net fluxes between the underlying states become irreversible, establishing an arrow of time [7,8]. Remarkably, the reversibility signature can reflect the complexity of the brain's functional organization and has been shown to relate to hierarchical processing in cases of altered states of consciousness [3,4,9]. It has been proposed that the brain processing is shaped by a hierarchy from unimodal to transmodal areas [10,11]. The link between hierarchy and signal reversibility has been explored in a previous study [1] showing how non-reversibility was used to estimate the level of orchestration changing according to the extrinsic driving of the environment. Nevertheless, most previous studies are based on human recordings (mainly using fMRI) and thus, lack the temporal resolution to assess non-reversibility in higher frequencies such as gamma band activity. Furthermore, much of the fMRI literature on irreversibility focuses on characterization and comparison of different brain states but not the transitions between them, as we present in the current study.

To this end, signal irreversibility has not yet been assessed in data recorded in a ferret population. Here, we chose the ferret brain due to the cortical homologies to the human brain and the possibility of measuring local field potentials (LFPs) from the surface of multiple areas distributed across the posterior half of the cortex [12]. Such mesoscale data are an ideal choice to demonstrate the strength of the metric. Due to the difficulty of having such continuous recordings in humans, by using ferrets it is possible to assess the states transitions in long periods, providing another strength to this type of data. Furthermore, the obtained results can be

compared and validated with embedded dynamics that can be obtained by dimensionality reduction approaches and generative models of switching behavior. In this way, we propose an easy and fast implementable approach, comparable to already validated techniques which reduce the noise in the signal and remove the redundancy of high-dimensional neuroimaging recordings. Overall, these data pose a unique analytical challenge–combining high temporal resolution neural signals with tracking of much slower behavioral changes. Can the extracted time series metric, such as irreversibility, that is applied to the finely sampled neural recordings, reveal a precise transition between changing behaviors?

In previous studies, functional connectivity in ferrets has been studied using implanted micro-electrocorticographic (µECoG) arrays during transitions between brain states (such as sleep and awake) in behaving animals [12] and transitions between different depths of anesthesia [13]. There, the different brain states were classified by using electrophysiological signatures (spectral distributions across channels) of neural activity detected in the cortex [14–16]. Distinct brain states showed different patterns of cross-frequency phase-amplitude coupling and inter-electrode phase synchronization across diverse frequency bands [12]. µECoG allows simultaneous monitoring of several different functional systems, while enabling recordings from multiple areas within the same cortical systems [12,16]. Furthermore, ferrets present a relatively quick alternation between sleep stages, allowing a detailed study of these transitions [17–20]. In the current study, we inspected the patterns of large-scale cortical functional connectivity across time, as the animals go through distinct brain states.

We inferred a Hidden Markov Model (HMM) to represent unique brain networks of distinct activity and functional connectivity that repeat at different points in time [21–23]. HMM offers a probabilistic (generative) model that, through a single process of Bayesian inference, models the time series in a self-contained manner [21–24]. We investigated widely reported metrics (such as switching rate) of brain substate's transition dynamics in order to assess whether there is agreement with the results provided by the irreversibility metric.

We hypothesize that irreversibility reveals differences in the neural dynamics of different brain activity states. We showed that the awake brain had the highest irreversibility value. Simultaneously, the awake state also exhibited the highest switching rate and other complexity metrics in the embedded dynamics. Furthermore, the irreversibility inspection not only revealed a hierarchical organization between different behavioral states, but also functional hierarchies regarding brain region and frequency band. The irreversibility metric provided information about different brain states at both the macro and meso-scale, suggesting a functional hierarchical organization. These results shed light on how different approaches of thermodynamics and information theory can reveal hierarchical orchestration of ongoing dynamics.

## Methods

### Ethics statement

All experiments were approved by the independent Hamburg state authority for animal welfare (Behörde für Gesundheit und Verbraucherschutz Hamburg) and were performed in accordance with the guidelines of the German Animal Protection Law.

### Animals preparation and recordings

Data from adult female ferrets (*Mustela putorius furo*) were used for analyses carried out in this study. Detailed explanation of animals housing, implantation and recording are presented in a previous publication [12]. Neural activity was recorded over a large portion of the left cerebral hemisphere including visual, auditory, and parietal areas (more than 2 hours recording

per animal). Recordings were performed with a custom designed µECoG array consisting of 64 equidistantly spaced electrodes (1.5 mm interelectrode distance) of 250 um diameter. During the ECoG implantation [12] the position of the µECoG on the cortex was photographed and offline projected onto a scaled illustration of a ferret brain map [25]. Data from each electrode were then allocated to the cortical area directly underlying the corresponding ECoG contact. After recovery from implantation surgery, ferrets were gradually accustomed to a recording box (45×20×52 cm) that was placed in a dark sound attenuated chamber where the animal was able to move freely. To monitor animal movement, an accelerometer was tightly attached to the cable-interface close to the head. µECoG signals were digitized at 1.4 kHz (0.1Hz high pass and 357 Hz low pass filters) and sampled simultaneously with a 64 channel AlphaLab SnRTM recording system (Alpha Omega Engineering, Israel). For further details on the animal implantation and recordings procedures see (Stitt et al., 2017) [12].

## Irreversibility

The irreversibility level was calculated on the µECoG signal of all the channels. Irreversibility was measured within non-overlapping sliding windows (of 1 second) analysis through the data [26], creating two different, forward and time-reversed backward, correlation patterns (the difference of which generates the irreversibility matrices) (**Fig 1B**). In other words, both forward and backward correlations were generated at every one second time-window, by calculating the shifted Pearson correlation in the corresponding direction of the signal. The difference between these two patterns at each sliding window provided the irreversibility magnitude at the corresponding time point [26]. The calculation of the metrics is described briefly below, for more information, see [1].

The assumed causal dependency between the time series $x(t)$ and $y(t)$ (two signals obtained from two different electrodes) is measured through the time-shifted correlation using a sampling rate (fs) of 100 Hz. For the forward evolution the time-shifted correlation is given by

$$C_{forward}(\Delta t) = <x(t), y(t + \Delta t)> \tag{1}$$

And for the reversed backward evolution the time-shifted correlation is given by

$$C_{reversal}(\Delta t) = <x^{(r)}(t), y^r(t + \Delta t)> \tag{2}$$

being $\Delta t$ = 1/fs based on previous literature [1]. This amounts to the minimum step size (one frame for the window) within which irreversibility is calculated taking away the arbitrary selection of a time window duration. Furthermore, the arithmetic mean (represented in the formulas as $<>$ can be calculated by

$$\text{Arithmetic Mean} = \frac{\sum_i^n x_i}{n} \tag{3}$$

Where the reversed backward version of $x(t)$ (or $y(t)$), that we call $x^{(r)}(t)$ (or $y^{(r)}(t)$), is obtained by flipping the time ordering.

The pairwise level of non-reversibility is given consequently by the absolute difference between the assumed causal relationship between these two timeseries in the forward and reversed backward evolution, at a given shift $\Delta t$ = 1/$ts$. For the current study the shifting was selected at T = 1.

$$I_{x,y}(T) = |C_{forward}(T) - C_{reversal}(T)| \tag{4}$$

Therefore, the level of irreversibility relies on the idea of finding the arrow of time through

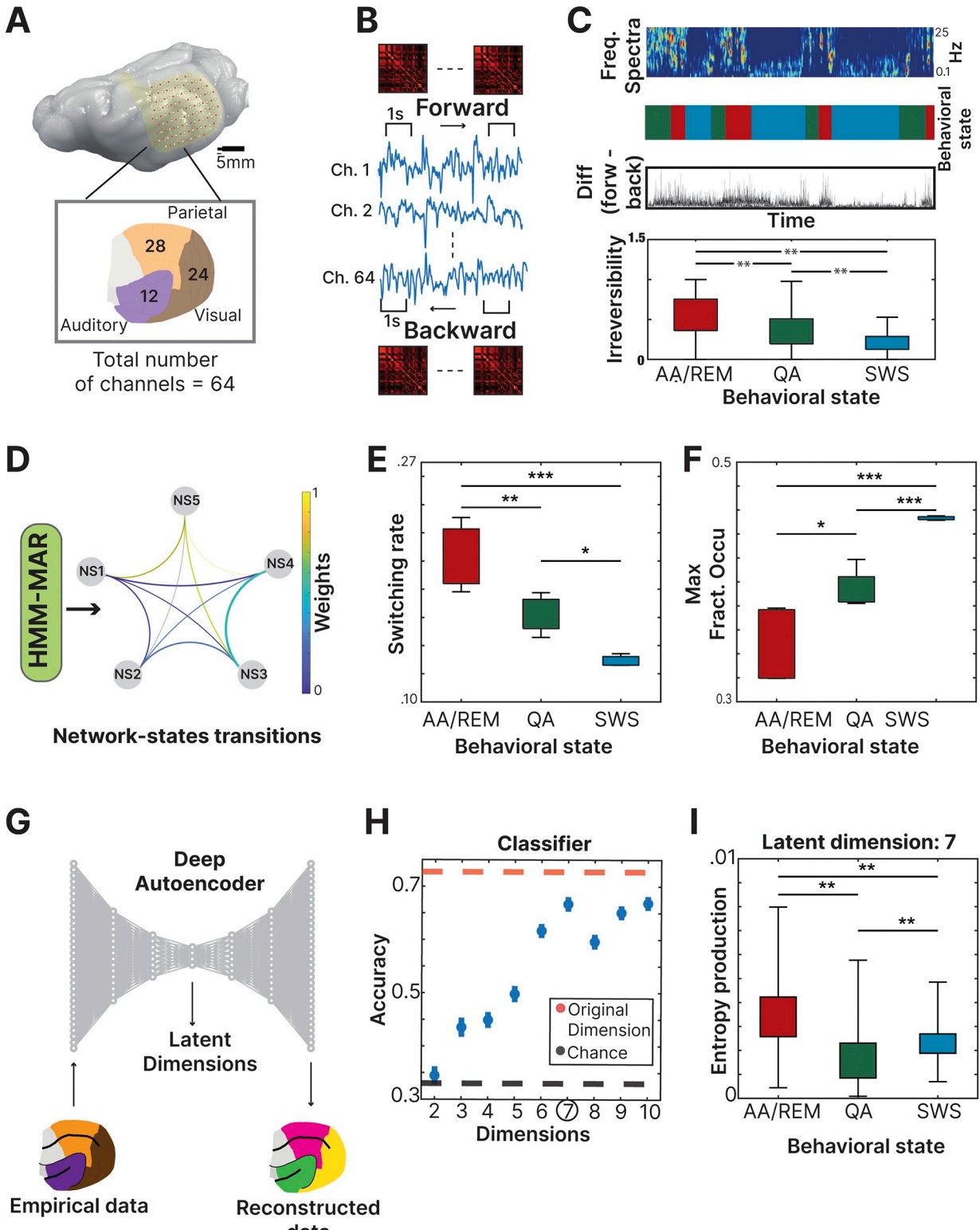

**Fig 1.** Analysis summary: **(A)** Data was recorded from 64 custom micro-electrocorticographic (μECoG) electrodes placed in the ferret cortex. The number of electrodes in each cortical functional system (auditory, visual, parietal) is depicted in the insert, which sum to 64 total electrodes. **(B)** Raw signal level of irreversibility was obtained by calculating for each time window (1 second) the time-shifted forward correlation difference with the time-shifted backward correlation. **(C)** Frequency spectra, classification of behavioral states and irreversibility values (difference between forward and backward matrix) at each time point across the whole recording (> 2 hours). Irreversibility value was calculated for each behavioral state

showing the AA/REM as the one with the highest level. **(D)** PCA was calculated for each brain region and the three first components (one from each area) were used as the input signal for the Hidden Markov Model Analysis (HMM). The HMM resulting network states are presented with their corresponding transitions. **(E)** The switching rate between the obtained network states was grouped by behavioral states, showing the AA/REM as the state with the highest switching rate (AA/REM: 0.15; QA: 0.13; SWS:0.12). **(F)** The maximal fractional occupancy of the 5 network states was calculated for each behavioral state, showing the SWS as the one with the highest value (AA/REM: 0.32; QA: 0.37; SWS:0.45). **(G)** We applied a deep autoencoder to reduce the dimensionality of the source data in order to explore the embedded dynamics of the system. **(H)** The distinction between the three behavioral states was assessed through a classifier at each reduced dimension, showing the highest level of performance at dimension 7. **(I)** The entropy production of each behavioral state is displayed at dimension 7 revealing that the greatest departure from equilibrium occurs in the stage AA/REM ($[F(2, 1787) = 390, p < .01]$). The highest entropy production was at the AA/REM state (mean = 0.0034, std = 0.0012), followed by the SWS state (mean = 0.0023, std = 0.001) and QA state (mean = 0.0017, std = 0.0011).

the degree of asymmetry obtained by comparing the lagged correlation between pairwise time series.

The extent to which the forward and reversed time series are distinguishable determines the reversibility/equilibrium level. Thus, when the forward and reversed time series are not distinguishable, the system is reversible and in equilibrium, whereas when the level of distinguishability increases, the system becomes more irreversible and away from the equilibrium.

## Hidden Markov Model

The Hidden Markov Model (HMM) is a statistical model that relies on the assumption that the signal can be well characterized as a parametric random process, and that the parameters of the stochastic process can be determined (estimated) in a precise, well-defined manner [27]. It describes a time series as a sequence of states, where each state has its own model of the observed data (i.e., the observation model) [23]. For this study, we used the multivariate autoregressive form of the HMM (HMM-MAR) as extensively described in previous literature [28–30]. Briefly, the MAR model characterizes the behavior of time series by linear historical interaction between the observed time series from different brain areas. MARs are able to characterize the frequency structure of the data, and by making the model multivariate, are able to capture interactions (e.g., coherence) between multiple brain regions [23]. As input for the model, we performed a principal component analysis (PCA) for the data of each brain region and used the first component from each area. In order to attenuate the effect of possible spatial leakage (due to the adjacency of neighboring areas), symmetric multivariate leakage correction was applied across the whole network as proposed by the creators of the used toolbox [24,31]. The model optimal output was found to be at 5 network states as it showed no predominant presence of one network state over the others exposing a heterogeneous distribution of them. For the outcome of the model, three metrics are reported in our result section. We selected these metrics as they have been reported extensively in previous literature and suggested as metrics of analytical interest by the toolbox used [22–24].

## Switching rate

Switching rate is defined as the metric of the algorithm that can be interpreted as the probability of switching between different substates [32] and it represents the stability of the dynamics [33].

## Maximal fractional occupancy

The fractional occupancy reveals the proportion of time points for which the HMM–MAR was in each particular state [23]. In other words, fractional occupancy is defined as the fraction of

time spent in each state:

$$Fractional\ occupancy(k) = \frac{1}{T}\sum\nolimits_t (u_t == k) \qquad (5)$$

Where, in the state space dimension K, the most probable a posteriori state ($u_t$) is one if $u_t = k$ and is zero otherwise, and T is the length of the network state sequence in samples [22].

We display the maximal value of this metric to reveal the preference of one substrate above the others in the different behavioral states as reported in previous literature [34].

## Average lifetime

The average lifetime represents the mean duration of the sub-states visit. In the scenario that the amount of time spent is the same for each sub-state, the average life-time would be identical for all of them [35].

## Irreversibility at different frequencies

We applied a bandpass filter of 1Hz in the range of 0 to 50 Hz resulting in 50 different filtered signals. We calculated the irreversibility metric to each of these signals and displayed the results (**Fig 2B**) indicating the average value of irreversibility by frequency band: Delta (1–3 Hz), Theta (4–6 Hz), Alpha (8–16 Hz), Beta (17–30 Hz) and Gamma (30–50 Hz) as reported in a previous study [12]. We will explore the influence of each frequency band to assess the driving effect of each one of them and to assess if there is a bias towards higher frequencies as reported previously [3].

## Information theory metrics

**Entropy production.** To perform basic biological functions such as information processing, organisms need to break detailed balance [2]. For a system that obeys detail balance, the probability of observing a transition between any given two states in the configuration space is the same as the probability of observing the reverse transition. Such a regime is called thermodynamic equilibrium. However, in systems far from equilibrium, the transition probabilities are asymmetric, and the detailed balance is broken, giving rise to temporally irreversible dynamics. Entropy production is a central concept of nonequilibrium statistical mechanics used to quantify the extent of this violation [36,37]. To define this quantity, we first consider a system with a set of joint transition probabilities $P(i{\rightarrow}j) \equiv P(x_t = i, x_{t+1} = j)$. We note that $P(i{\rightarrow}j)$ differs from the conditional transition probability $P(j|i) = P(x_{t+1} = j | x_t = i)$. In particular, for a Markovian system with a stationary distribution $\pi$, both quantities are related by $P(i{\rightarrow}j) = P(j|i)\pi(i)$. In such systems, an information-theoretic form of entropy production is obtained by computing the Kullback-Leibler distance or relative entropy between forward transition probabilities $P(i{\rightarrow}j)$ and the reverse transition probabilities $P(j{\rightarrow}i)$

$$S = \sum\nolimits_{i,j} P(i \rightarrow j) \log \frac{P(i \rightarrow j)}{P(j \rightarrow i)} \qquad (6)$$

This distance measures the departure from detailed balance by comparing the probabilities of the observed process and its time-reversed version. If the system satisfies the detailed balance condition, the joint transition probabilities between any two states are equal to the reverse probabilities, $P(i{\rightarrow}j) = P(j{\rightarrow}i)$, making all arguments of the logarithms in (6) equal to 1, and consequently leading to null entropy production. On the other hand, any deviation from detailed balance, $P(i{\rightarrow}j) \neq P(j{\rightarrow}i)$, makes the transition probabilities asymmetric, resulting in a positive increase in entropy production.

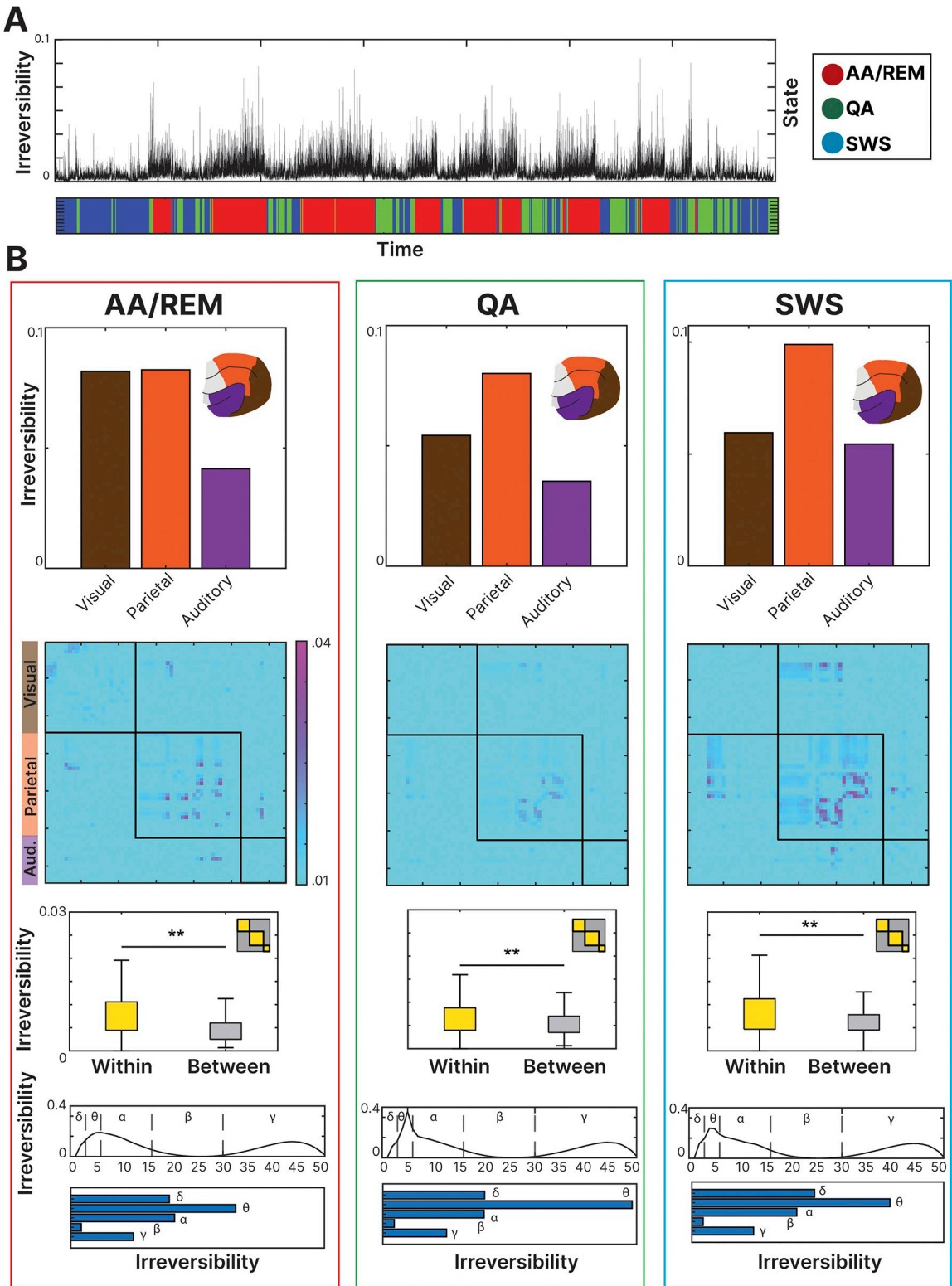

**Fig 2.** Irreversibility level across time and regions: **(A)** The irreversibility level was assessed at each time point, displayed next to the behavioral state at the corresponding moment. Over time, a significant increase in irreversibility can be observed during the Active-Awake(AA)/REM stages. **(B)** Signal irreversibility was calculated for the three different behavioral states. AA/REM (left), Quiet-Awake (QA) (center) and Slow-wave-sleep (SWS) (right). For each state, it is displayed the level of irreversibility for each functional system (Visual, auditory, and parietal) and the level corresponding to the within-system relations and the between-system relations. Across

sleep stages, the parietal cortex and within-system relations were revealed to be the drivers of the irreversibility in the system. At the bottom, the irreversibility level was measured at bins of 1 Hz between 0 and 50 Hz. The analysis was also performed for all the different behavioral states. Bar figures indicated the level of irreversibility when grouping the frequencies by previously reported frequency bands being the theta range the one with the highest irreversibility value.

In certain situations, it is meaningful to compute a pairwise form of entropy production. In a system composed of multiple interacting elements, the deviation from the equilibrium due to the interaction of two elements $a$, $b$ is given by:

$$S_{ab} = \sum_{i,j} P_{ab}(i \rightarrow j) log \frac{P_{ab}(i \rightarrow j)}{P_{ab}(j \rightarrow i)} \tag{7}$$

where $P_{ab}(i{\rightarrow}j)$ represents the forward transition probabilities of the subsystem composed only by the elements i and j, and $P_{ab}(j{\rightarrow}i)$ the backward transition probabilities. We refer to this metric as pairwise entropy production. In previous works, pairwise entropy production was shown to capture most of the irreversibility of neural data, with respect to higher-order entropies [38]. Taking this into account, we tested the hypothesis whether the different stages could be classified using pairwise entropies $S_{ab}$ alone.

## Determinism

Determinism measures how much information a system's state transition graph encodes about its future evolution. The determinism is low in a system where each state has an equiprobable chance of evolving into one of many future states, while a system where each state evolves with probability equal to 1 to a subsequent state would be highly deterministic [39,40]. The average determinism of a directed network $x$ with N (amount of states in the transition probability matrix) can be quantified as

$$Det(x) = \frac{log_2 N - <H(W_i^{out})>}{log_2 N} \tag{8}$$

where $< H(W_i^{out}) >$ corresponds to the average entropy of the probability distribution of possible futures (weighted out-going edges) for each node $i$ [40].

## Degeneracy

Degeneracy gives a measure of how much information a system's state transition graph encodes about its past evolution. A system where all states feed into the same future would be described as highly degenerate, while a system where each state had a well-defined past would exhibit low degeneracy [39,40]. Degeneracy is calculated as

$$Deg(x) = \frac{log_2 N - H < (W_i^{out}) >}{log_2 N} \tag{9}$$

Where the $(W_i^{out})$ is a vector in which every component contains the mean probability of transition of each state. Therefore, the entropy is calculated over the distribution defined by the components of the vector.

## Mutual information

The calculation of mutual information (MI) quantifies how much knowledge of the past state of the system reduces our uncertainty about the future state of the system [41]. MI is one of the many approaches to measure independence between different variables [42]. MI is zero if and

only if, the two random variables are independent. MI's theoretical advantages are unique in its close ties to Shannon entropy [43]. The quantification of mutual information exposes information between the future and past joint states of the variables that make up a network. Therefore, as a result, mutual information is calculated by adding the information (entropy) of the previous ($H(X)$) and future ($H(Y)$) state and subtracting the joint entropy ($H(X,Y)$) of them. The equation is as follows:

$$MI(X; Y) = H(X) + H(Y) - H(X, Y) \tag{10}$$

## Dimensionality reduction and classification of stages

Here we applied a similar methodology used in a previous publication [12] to classify recordings into brain stages. Briefly, a spectrogram (sliding window of 60 s in steps of 5s, 14 frequencies logarithmically distributed between 0 and 128 Hz) was constructed individually for each of the 64 channels and z-scored in the frequency axis to equilibrate the weight across frequencies. Posteriorly, all 64 spectrograms were averaged into a single global spectrogram. Epochs with mean z-score higher than 10 were considered as noise epochs and consequently data were rejected in a window of ± 10 seconds from all time points that exceeded this threshold. By means of PCA (applied on the entire 64 channel data), the dimensionality was reduced to the 8 strongest components on which a k-means cluster analysis was applied to classify each recording session into 3 clusters. Based on the physiological characteristics, the stages were classified as Active-Awake(AA)/REM, Quiet-Awake (QA), and SWS.

## Deep autoencoders

Autoencoders are deep artificial neural networks that are used to reduce the dimensionality of data. Specifically, an autoencoder is a feedforward network that is trained to output an approximate reconstruction of its input while learning a low-dimensional representation of the data. Unlike other methods such as PCA (or derivatives of it), autoencoders reduce the dimensionality by nonlinear coordinate transformations. Autoencoder networks consist of two parts: the encoder, which compresses the data into a lower dimensional space (called latent space) and the decoder which reconstructs the data in the original dimension [44–47].

In this work, we implemented an architecture of 6 fully connected layers (64, 32, 16, **d**, 16, 32, 64). The dimension of the latent space "**d**" was chosen as small as possible, as the minimum number of units that preserves the relevant information of the arrow of time in the data (see Results). The input of the network was an array of 64 dimensions representing the state of each channel at a certain timepoint. We used batch normalization and rectified linear unit (ReLU) activation functions in all layers, except the middle and last one which acted as linear units. The minimization of the mean square error (MSE) between the input data and the output layer was used to train the network, through the Adam optimizer [48]. The data was previously z-scored and split into a training and test set (70–30% of the dataset). The batch size was 256 timepoints and the networks were trained for 400 epochs, or until the validation error was no longer improving for the last 50 epochs.

## Classification of stages in the low dimensional space

To investigate the minimal number of latent dimensions needed to preserve information on the arrow of time, we trained classifiers to distinguish stages in the low dimensional space. In this way, we intended to assess if signal irreversibility provided enough information to disentangle the different brain states in lower dimensions.

For each d-dimensional representation of the data, we binarized the timeseries (with a cut-off value of zero) and computed the pairwise entropies $S^{(ij)}$ between all pairs of encoded coordinates, as described previously in the methods section. The irreversibility was computed in windows of 5 seconds (2500 points with 500Hz of sampling rate). Thus, we obtained arrays of entropies for each encoding model that we used as features for the classification. As the signal got binarized, we intended to prove that having only two "Micro-states" (0 and 1) it is already enough to achieve our goal of entropy calculation.

We implemented a random forest classifier using the Scikit-learn library in Python as performed in previous studies [49,50]. As the dataset was imbalanced (the number of windows corresponding to the three stages was not the same), we applied under-sampling without replacement to achieve the same number of elements of each class. The under-sampling process was repeated 50 times. To achieve statistical significance, we used leave-one-out cross validation, splitting the set into train (80%) and test (20%) 50 times for each sampling.

### Statistics

We performed paired statistical analyses (t-test/ANOVA, depending on the group number) as the comparison was performed for the same animal in different brain states. Animal ID was added as a random effect to control the violation of independence of the data. Statistical significance was assessed with a threshold of p = .05. All *post hoc* comparisons were Bonferroni corrected for the number of comparisons taking place. The code for obtaining the metrics and estimation of the results can be accessed in the provided link.

## Results

To investigate the functional organization of large-scale dynamics, we analyzed electrophysiological information obtained from recordings performed in ferrets. We obtained data from 4 animals in resting activity during the light cycle between 12 a.m. and 6 p.m (more than 2 hours recording per animal). We recorded LFPs by using 64 custom micro-electrocorticographic (μECoG) electrodes displayed in arrays chronically implanted in the animals (**Fig 1A**). To disentangle the level of temporal irreversibility, we used the electrocorticographic (μECoG) time series obtained during spontaneous behavior, including movement, quiet-awake, and sleep periods. In this study, we replicated results reported in previous articles [12] using the same dataset in order to validate the presented results, showing that the highest recording time was in the slow wave sleep state and that the most probable state transition was from active-awake/REM to quiet-awake (**S1A and S1B Fig**). Furthermore, we compared the power spectrum of each corresponding state as presented in [12] (**S1C Fig**).

We looked at the brain dynamics of behavioral states transition through electrophysiological data. Benefitting from the ferrets' shorter duration of states, we inspected the temporal asymmetry of the data by extracting both the forward time series and its reverse version. By comparing the difference between these two, we were able to extract the level of non-reversibility of the system. Along these lines, irreversible macroscopic dynamics have been treated as a signature of non-equilibrium processes occurring at the micro-scale in biological systems [2, 51]. This approach sheds light onto the brain dynamics and hierarchy at the different brain states.

We evaluated the shifted correlation in both directions (forward and backward) in a sliding time window of 1 second (**Fig 1B**). The absolute difference between the irreversibility matrices (forward minus backward) displays the irreversibility value for each of the time points (**Fig 1C**). In other words, the irreversibility metric was calculated over all channels (by assessing the difference in the corresponding matrices, see methods), providing one unique reversibility

value for each time point. The frequency spectrum was used to determine the behavioral state of the animal across the whole recording time (**Fig 1C**). Using a mixed-effect ANOVA (See statistics section), we found that there was a significant difference in the irreversibility level between all the behavioral states ($[F(1, 3660892) = 3928, p < .01]$ being "Active-awake/REM (AA/REM)" sleep the state with the highest value of irreversibility (mean = 0.05 +/- 0.002), followed by "Quiet-awake (QA)" (mean = 0.03 +/- 0.002) and "Slow wave sleep (SWS)" (mean = 0.02 +/- 0.001) (Post-hoc Bonferroni corrected while all comparison where significant: $p < .01$) (**Fig 1C**). We incorporated complementary approaches in order to analyze the embedded dynamics of the system and their differences in the different brain states. With that goal, and to add robustness to the obtained results, we inspected recurring activity patterns across time by applying HMM in order to observe, in a different way, the embedded dynamics in the signal. For that goal, we calculated a PCA of each brain region (see Methods) and used the first component from each area as input for HMM. The HMM analysis resulted in 5 network states (**Fig 1D**). We defined the number of network states after an exploration of different value between the range of 4 to 7 which indicate 5 as the optimal [29]. We inspected the properties of the network states by comparing their dynamics in the different behavioral states.

By means of a mixed effect ANOVA (see statistics section), we observed a significant difference between the three behavioral states ($[F(1, 10) = 26.47, p < .01]$) where AA/REM had the highest level of switching rate (0.155), followed by QA (0.133) and SWS (0.124) (**Fig 1E**). There was a significant difference in all the comparisons between the conditions (AA/REM-QA: $p < .001$, AA/REM-SW: $p < .01$, QA-SWS: $p < .05$). The reversed pattern was observed when looking into the maximal fractional occupancy ("MaxFO", the highest value of time staying at each network state). There was a significant difference between the three behavioral states ($[F(1, 10) = 170.15, p < .01]$) where SWS displayed the highest MaxFO (0.44), followed by QA (0.37) and then (0.32) (**Fig 1F**). There was a significant difference in all the comparisons between the conditions (AA/REM-QA: $p < .05$, AA/REM-SW: $p < .001$, QA-SWS: $p < .001$).

We used a deep autoencoder to reduce the dimensionality of the signal and inspect the latent information (**Fig 1G**). The pairwise entropy production in the reduced space was shown to be sufficient to classify different stages, achieving an accuracy of 0.67 when using only 7 latent dimensions, comparable with the performance of the model using the pairwise entropies of the raw data (consisting of 64 channels). Adding more dimensions did not result in a significant improvement in the accuracy, showing that 7 dimensions are enough for capturing the relevant information on the arrow of time that characterizes the stages (**Fig 1H**). When computing the total entropy production in this dimensionality, there was a significant difference ($[F(1, 1788) = 2867, p < .01]$) where the AA/REM showed the highest level of entropy, as observed previously in the switching rate of network-states of the HMM and in the irreversibility analysis at the source space (**Fig 1I**). There was a significant difference in all the comparisons between the conditions (AA/REM-QA: $p < .01$, AA/REM-SW: $p < .01$, QA-SWS: $p < .01$). It is relevant to clarify that even if the AA/REM state revealed to be the predominant state in both metrics (irreversibility in the source space and entropy production in the latent space), the order between the remaining two states is inconsistent. Further studies should dive into this difference to understand this phenomenon.

In summary, AA/REM showed to have the highest level of irreversibility between all the possible states. Accordingly, it also revealed to have the highest switching rate between the network-states (derived from the HMM), the lowest value of fractional occupancy and the highest entropy production (derived from the dimensionality reduction). These results probed how metrics of non-equilibrium contribute to the disentanglement of the different brain states.

## Irreversibility reveals brain hierarchy for different states and cortical systems

The information obtained through the irreversibility analysis showed how the different behavioral states vary in their dynamics. Here, we explore these dynamics in more depth by taking into account the functional structure of the brain.

We inspected the level of irreversibility for all the electrodes across time with the corresponding behavioral state at each time point (**Fig 2A**). For each state, we performed the same analysis but segmented for the different cortical functional systems (auditory, visual, and parietal). The recording electrodes were grouped into either the occipital (brown), parietal (orange), or temporal (purple) functional system based on the cortical area that was underlying each electrode (**Fig 2B**). We compared the level of irreversibility observed at each system (top). The level of irreversibility was higher in parietal cortex in all the states (AA/REM: .0083, QA: .0080, SWS: .0097) compared to visual (AA/REM: .0082, QA: .0054, SWS: .0060) and auditory (AA/REM: .0041, QA: .0035, SWS: .0054). The difference was significant in all these comparisons: AA/REM ($F(1, 1500) = 60.68, p < .01$), QA ($F(1, 1500) = 51.82, p < .01$) and SWS ($F(1, 1500) = 52.76, p < .01$). This suggests differences in the processing dynamics between higher-order parietal regions and sensory areas, compatible with the notion of functional hierarchies. Irreversibility provided information about this distinction and allowed further exploration of the comparison between different cortical subnetworks.

Given these proposed organizational principles, we set out to investigate how the brain balances the irreversibility of local within-system relations (e.g., intra-system edges; on-diagonal black blocks in **Fig 2B**), and long range between-system relations across brain states. For within-system relations we combined signals from μECoG recording electrodes that were positioned over occipital, parietal, and temporal cortical systems, respectively (**Fig 2B**) while between-system relations consisted of the signal combinations between the remaining areas. In other words, we compared the level of irreversibility of the within-system relations (each area with itself—inside black squares in center figure) against the level of the between-system sections (each area with the others- outside black squares in center figure) (**Fig 2B**). The observed pattern was consistent disregarding the different number of electrodes per system (**Fig 1a**). By means of paired t-student comparisons, we found a significant difference between the within-regions (connections between brain areas of the same regions) and between-regions (connections between brain areas of the different regions) irreversibility values ($p < .01$). The difference was significant at AA/REM state ($t(4095) = 215.30, p < .01$), QA ($t(4095) = 215.73, p < .01$) and SWS ($t(4287) = 225.86, p < .01$). The level of irreversibility was higher in the within-system relations in all the states (AA/REM: .0069; QA: .0057; SWS: .0074) compared to the between-system relations (AA/REM: .0045; QA: .0052; SWS: .0063). It is important to clarify that existing literature has already proven how the cross-correlation of neighboring cortical areas is a genuine feature of the data and does not reflect statistical noise. In this way, the cross-correlation and spatial leakage (see methods) would not be potential driving factor for the estimation of the signal irreversibility [52].

Lastly, we inspected the irreversibility of the system at different frequency bands. For that goal, we obtained the irreversibility value at each frequency bin (1Hz) and compared the results divided on the previously reported bands. We observed that the predominant frequency band was the theta band (3–7 Hz) showing a significant difference from the remaining bands ($F(4, 44) = 33.18, p < .01$) while the least prominent was the beta band (17–30 Hz) (**Fig 2B**).

In summary, the irreversibility of the signal not only provided information about the different states and their transition, but also about the different roles of distinct brain areas and frequencies over the non-equilibrium of the signal. The parietal system demonstrated the highest

influence on the dynamic hierarchy (by having the highest irreversibility levels of any system) while generally, within-system relations were shown to have a higher value of irreversibility than the between-system relations.

## Embedded network system revealed different entropy dynamics according to the behavioral state

The behavioral states dynamics can be inspected through the observation of the transition between them. We revisit the results obtained in Fig 1D to extend the analysis. In this section we aim to reconnect the results presented previously with several metrics obtained through the observation of the transitions across the different network states.

Using the HMM framework, we obtained 5 network substates and inspected information theory metrics in order to study if the dynamics between the substates are consistent with the previously reported results.

As an input to the HMM, we used the first principal component of each functional system (**Fig 3A**). The resulting 5 network states were displayed with their corresponding probability at each time point, while the predominant state (network state with the highest probability at that moment) is displayed below (**Fig 3B**). The average amount of time spent at each network state is shown for each behavioral state (**Fig 3B**). From this visualization, we observe that during SWS, the HMM disproportionally labeled that time as network state 3. We calculated the entropy production of the system by comparing the temporal symmetry between the different network states. If the relation between the network states was the same in both directions (symmetrical), then the entropy production level would be 0. The more different the relations were (asymmetrical), the higher the entropy production value (**Fig 3D**). There was a significant difference between the different states ($[F(1, 10) = 5.73, p < .05]$) being the difference between AA/REM and SWS, the highest one. The AA/REM had the highest entropy production level (mean = 0.0135, std = 0.0079), followed by QA (mean = 0.0118, std = 0.0075) and lastly SWS (mean = 0.0087, std = 0.0054) (**Fig 3E**). The only significant difference between the conditions was observed in the AA/REM-SWS comparison ($p < .05$). The transition probability matrix of each behavioral state is displayed next to the corresponding difference matrix (upper triangle minus lower triangle) (**Fig 3F**). The difference between the behavioral states was also assessed using three different metrics of information theory resulting in the same pattern where AWREM showed the lower value, followed by AA/REM and SWS: determinism (AA/REM: .54; QA: .55; SWS: .57) (**Fig 3G**), degeneracy (AA/REM: .11; QA: .15; SWS: .22) (**Fig 3H**) and mutual information (AA/REM: 1.30; QA: 1.42; SWS: 1.57) (**Fig 3I**). There were no significant differences neither in determinism ($[F(1, 10) = 0.06, p = .79]$), nor degeneracy ($[F(1, 10) = 1.08, p = .32]$), nor mutual information ($[F(1, 10) = 0.01, p = .89)$ (multiple comparisons were not significant and addressed via Bonferroni correction). Despite showing a consistent trend, in all the information theory metrics (**Fig 3G, 3H, and 3I**), the difference between the states was not significant ($p > .13$) mostly due to the low number of animals considered for the analysis. An extra analysis, through a Bayesian ANOVA (https://github.com/klabhub/bayesFactor) showed a high level of certainty in the three metrics confirming the support for the null hypothesis (Certainty: Determinism 84%, Degeneracy 90%, Mutual information: 84%).

In summary, the results obtained from the HMM inspection revealed a consistent pattern with the outcome of the irreversibility analysis. AA/REM showed the highest entropy production while the lowest level in determinism, degeneracy, and mutual information. These results revealed how the different behavioral states are disentangled using information theory metrics consistently to the previously reported thermodynamic features. By incorporating these metrics, we show converging evidence that this phenomenon is supported by several lines of

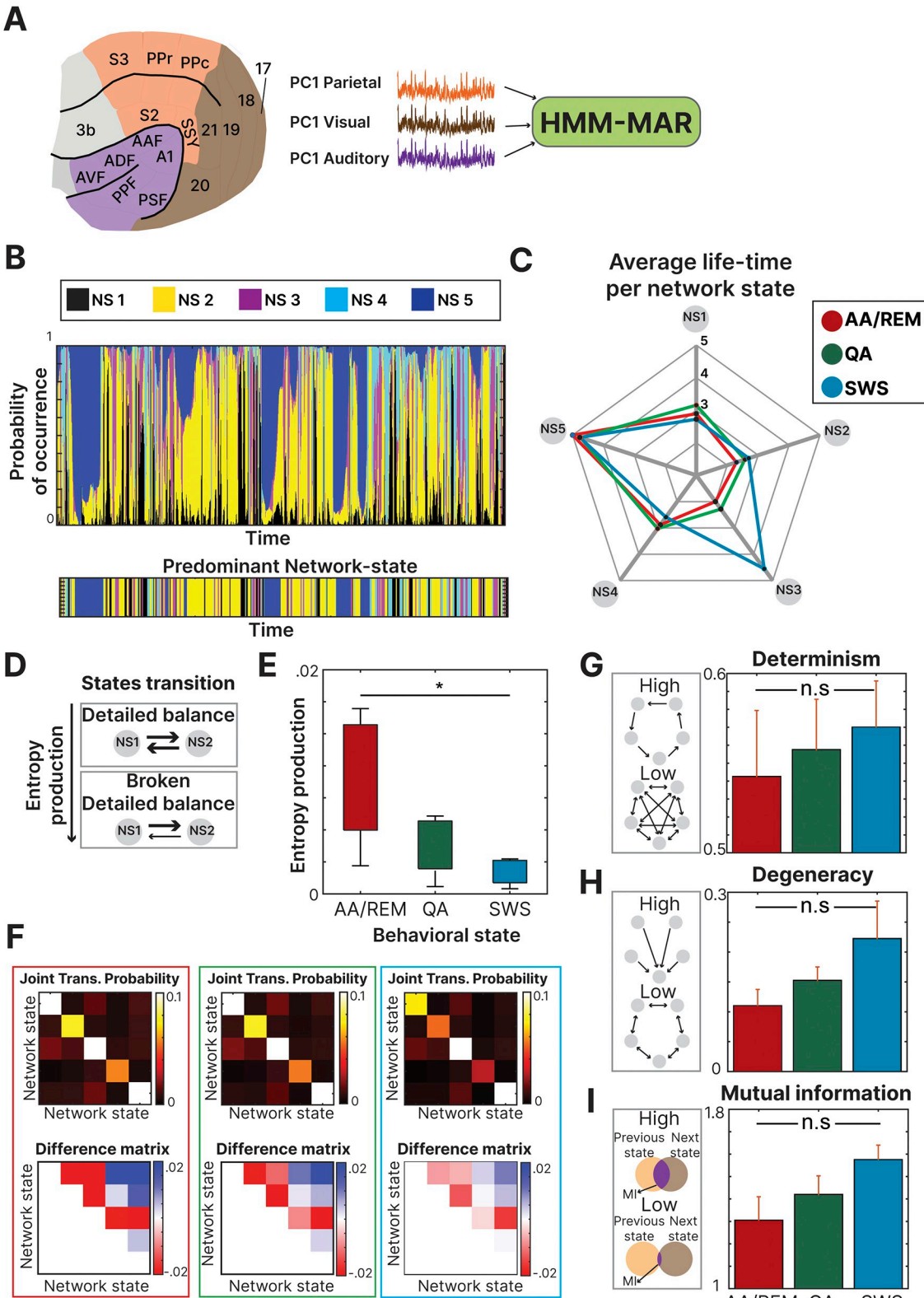

**Fig 3.** HMM analysis: (**A**) Principal component analysis was calculated for each functional system and the three first components (one from each system) were used as the input signal for the Hidden Markov Model Analysis (HMM). (**B**) At each time point, the probability of occurrence of each network-state is displayed, next to the predominant one for the corresponding moment. (**C**) The average life-time (time spent at each network state) was calculated for each behavioral state and displayed together to observe the similarities between

them (**D**) Entropy production was calculated by observing how asymmetrical the transitions between the network-states were. (**E**) The results, grouped by behavioral state indicated a higher level of transfer entropy in the AA/REM state (**F**) The joint transition probability between the network states is displayed for each behavioral state. Furthermore, the difference matrix (upper triangle minus lower triangle) is displayed to enhance the visualization of the effect (**G**) The level of determinism was calculated showing the highest value for the SWS state (**H**) The level of degeneracy was calculated showing the highest value for the SWS state (**I**) The level of mutual information was calculated showing the highest value for the SWS state.

evidence. Through coherent results obtained in the distinct metrics, the underlying brain dynamics can be explored in order to shed light on this topic.

## Discussion

We used electrophysiological data to study functional dynamics of cortical areas, involving visual, auditory, and parietal regions. We inspected the irreversibility of these neural signals and used this feature to label different behavioral states. Furthermore, results obtained from HMM analysis were in agreement with the observations made in the irreversibility analysis.

We measured the irreversibility of the data and implemented it as a proxy of the degree of non-equilibrium. Furthermore, this approach can be used in a manner to localize the differential influences of various cortical subsystems on this phenomenon. Its application contributes to enriching the knowledge of the brain's organization and could potentially serve as a future biomarker.

Previous studies have inspected non-equilibrium of signals in macaque monkeys [1,4], allowing to disentangle different consciousness levels through the non-reversibility of the signal. Nevertheless, the distinction between the brain states was due to pharmacological intervention instead of occurring spontaneously as in the present paper. Our study benefits from the relatively short duration of behavioral states in the ferret model and the extensive recording time (>3 hours per animal) in order to study the transition between the different states. Besides the significant difference in the recording durations, our study consists of continuous recordings from the same animals, allowing the exploration of the transition between the states, in contrast to previous studies, where the recordings of the different states were discrete and recorded in different periods of time.

A previous work has proposed that increases in irreversibility relate to changes in hierarchy levels, with low irreversibility signaling a flatter hierarchy in non-wakeful states [1,10]. We expanded our inspection on functional hierarchies in brain dynamics by comparing the irreversibility of the signal at different brain regions. We found a difference in irreversibility level in the distinct functional systems, that was consistent across all behavioral states, in which the parietal cortex appears showed the highest degree of irreversibility. Furthermore, within-system edges [53,54] were shown to be more irreversible than the between-system relations, showing a specific organization of the brain dynamics. Also, while comparing the influence of different frequencies, we reported a clear predominance of theta oscillations, which could be driving this metric of non-equilibrium as the main influence for the transition between the behavioral states [55]. Surprisingly, even though there is a higher presence of autocorrelation in lower frequencies, this did not interfere with the predominant role of the theta oscillations in the brain state transition dynamics. This is the first study considering the influence of both brain localization and frequencies influence in the measure of brain hierarchy through the irreversibility of the system.

The relation between brain organization and HMM have been explored in previous studies [21,56] showing how networks dynamics were hierarchically organized in different sets of metastates. In this study, we presented evidence to link this phenomenon with the dynamics presented in the irreversibility analysis. With this goal, we showed that there is a similar pattern

between the metrics, as the SWS behavioral state exhibited a higher reversible level while simultaneously showing a lower entropy production, and the lowest switching rate. The obtained results demonstrate how a common underlying mechanism which is altered in different behavioral states as reported in previous literature [6,9] can be captured in diverse ways, including direct metrics, such as irreversibility or indirect calculations through dimensionality reduction, such as HMM or the use of autoencoders. Previous studies have found distinctive levels of entropy production at different awareness states through fMRI data [6,9] showing convergent evidence with the results presented in this study, disregarding of the used neuroimaging technique.

Further embedded information about brain dynamics could be provided via information theory, which can denote interactions, associations or even dependencies between different systems [39]. We applied several information-theoretic metrics to the network states obtained through the HMM analysis and observed a congruent trend (despite not being significant due to the low sample size) with previous literature comparing different levels of complexity according to the corresponding brain state [40]. This is the first study comparing such embedded metrics with irreversibility dynamics of neuroimaging data across different behavioral states and proving the similarity between both outcomes, showing that direct and indirect metrics converge to the same hidden dynamics.

Furthermore, the brain is a highly complex dynamical system able to produce a vast repertoire of activity patterns. However, in spite of all the possible neural patterns available, only a few types of temporal patterns are observed [57]. Therefore, it has been hypothesized that the collective dynamics of the underlying neural processes are low dimensional. Indeed, multiple experiments showed that the dimensionality (degrees of freedom, or the number of variables required to explain a fixed variance) of the neural data is much lower than the number of recorded units [58]. In consequence, the dimensionality of the system can be reduced to lower-dimensional space. The classification of the brain states at lower dimensions revealed how the metric of entropy could disentangle between different states in lower dimensions with accuracy similar to the original dimension. This result demonstrates the feasibility and effectiveness of using these dimensionality reduction techniques. Even though the dimensionality of the system was reduced by approximately an order of magnitude, similar dynamic patterns were observed. By presenting results obtained through HMM and autoencoder dimensionality reduction, we show convergent evidence that strengthens the hypothesis framed in this article. The alternative approaches show different features of the same underlying phenomena, providing robust techniques for its exploration. Furthermore, it shows the association between entropy production and time-irreversibility, as it has been previously proposed [59,60].

In contrast to similar methods, the irreversibility framework has the flexibility that allows it to be applied to many different modalities including fMRI and MEG [5,26,61]. As it can be easily adapted to other techniques, it allows for an easy comparison of different datasets with different modalities of similar samples of subjects, in humans or other animals. In addition, we can assess whether the levels of non-equilibrium change with disease and therefore could be used as a potentially sensitive and specific biomarker. The irreversibility approach provides an indirect proxy to the nonequilibrium phenomenon of the system. Other more direct ways to unveil a system's thermodynamics, coming from methods which find other relations between dynamical and thermodynamical system properties [62–64] could provide alternatives to the one proposed in this article [65].

We emphasize that the presented framework can be applied to any set of time-series data. Therefore, the methods could not only be applied to brain data, but also can be used broadly to investigate broken reversibility and equilibrium in other complex living systems [2]. One interesting future step would be to develop a deterministic dynamical system, able to generate time series with adjustable degrees of irreversibility, and whose irreversibility is provable, not

based on observational noise, and not detected by any of the tests here reviewed [65]. As a classification method for the behavioral states, we employed a data-driven approach based on the time-varying nature of cortical LFP spectral properties. Similar data-driven approaches in mice have achieved a state classification accuracy of over 90% [66]. Other criteria could be used in order to disentangle the different states with different levels of accuracy.

In conclusion, we demonstrated how metrics related to thermodynamics such as signal reversibility, which have been treated as a signature of non-equilibrium processes, reflect brain dynamics distinguishable between different behavioral states. Furthermore, we found similar results (coherent trends following the same pattern) captured by statistical models and information theory metrics. Furthermore, we expanded the approach of irreversibility by inspecting the dynamic hierarchy of the system across different brain subsystems and the frequency range with the highest influence. All the presented results strengthened the application of the irreversibility metric as a biomarker for different brain states and disorders.

## Supporting information

**S1 Fig.** Description of relative recording time, transition between states and predominant frequencies: **(A)** The percentage of the recording time indicates the SWS as the behavioral state with the highest percentage while **(B)** the transition probability matrix indicates the transition from AA/REM to QA as the one that occurred the most amount of times. **(C)** The power spectrum displayed the predominant frequencies for each of the behavioral states.
(TIF)

## Acknowledgments

We thank Melina Timplalexi for the help with the figures. Furthermore, we thank Diego Vidaurre and Laura Masaracchia for the contributing code and opinions.

## Author Contributions

**Conceptualization:** Sebastian Idesis, Edgar Galindo-Leon, Andreas K. Engel, Gustavo Deco.

**Data curation:** Florian Pieper, Edgar Galindo-Leon.

**Formal analysis:** Sebastian Idesis, Sebastián Geli, Joshua Faskowitz, Jakub Vohryzek, Yonatan Sanz Perl, Edgar Galindo-Leon.

**Methodology:** Sebastian Idesis, Sebastián Geli, Joshua Faskowitz, Jakub Vohryzek, Yonatan Sanz Perl.

**Resources:** Florian Pieper, Edgar Galindo-Leon, Andreas K. Engel, Gustavo Deco.

**Supervision:** Gustavo Deco.

**Visualization:** Sebastian Idesis, Jakub Vohryzek.

**Writing – original draft:** Sebastian Idesis.

**Writing – review & editing:** Sebastian Idesis, Sebastián Geli, Joshua Faskowitz, Jakub Vohryzek, Yonatan Sanz Perl, Edgar Galindo-Leon, Andreas K. Engel, Gustavo Deco.

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
