## [Decision Letter · Decision Letter 0]

12 Oct 2023

Dear Mr. Idesis,

Thank you very much for submitting your manuscript "Functional hierarchies in brain dynamics characterized by signal reversibility in ferret cortex" for consideration at PLOS Computational Biology.

As with all papers reviewed by the journal, your manuscript was reviewed by members of the editorial board and by several independent reviewers. In light of the reviews (below this email), we would like to invite the resubmission of a significantly-revised version that takes into account the reviewers' comments.

Dear authors,

Both reviewers find the topic of your manuscript interesting and timely. However, they express concerns regarding the methods: the details of the analysis appear to be insufficient to properly assess the claims made. I believe that once the statistical and methodological approach and how this leads to the conclusions is provided in more detail, this manuscript can be suitable for publication in PLOS Computational Biology.

Kind regards,

Fleur Zeldenrust

We cannot make any decision about publication until we have seen the revised manuscript and your response to the reviewers' comments. Your revised manuscript is also likely to be sent to reviewers for further evaluation.

Sincerely,

Fleur Zeldenrust

Academic Editor

PLOS Computational Biology

Lyle Graham

Section Editor

PLOS Computational Biology

Dear authors,

Both reviewers find the topic of your manuscript interesting and timely. However, they express concerns regarding the methods: the details of the analysis appear to be insufficient to properly assess the claims made. I believe that once the statistical and methodological approach and how this leads to the conclusions is provided in more detail, this manuscript can be suitable for publication in PLOS Computational Biology.

Kind regards,

Fleur Zeldenrust

Reviewer's Responses to Questions

**Comments to the Authors:**

Reviewer #1: In "Functional hierarchies in brain dynamics characterized by signal reversibility in ferret cortex", Idesis et al. describe an analysis of μECoG data in ferrets that was described previously in Stitt et al. (2017). In the current analysis, the authors compare temporal irreversibility across different behavioral states (i.e., active awake, quiet awake, asleep), and further relate it to different metrics of brain activity dynamics.

The authors provide interesting results on how temporal irreversibility, a proxy for broken detailed balance, is related to transient temporal dynamics (HMM) and brain region. My biggest concerns are in regards to missing methodological details, motivations for using specific parameters, methods, and comparisons, and interpretation. Please see my comments below.

Major concerns

1. In the HMM pipeline, the authors take the first principal component of each brain region. Firstly, they do not provide details on the different brain regions, nor how many electrodes are in each region – on a closer inspection, it appears this information is present in Figure 3. I suggest the authors restructure this figure such that this information can be found earlier in the paper.

Secondly, after the PCA step, it is unclear whether/how the authors deal with spatial leakage (zero-lag correlations) before fitting the HMM.

Third, the authors do not explain or motivate why they chose to look at the two summary metrics of the HMM: switching rate, and maximal fractional occupancy. Especially the choice of the latter is unclear to me, as is its definition.

Fourth, are the differences in the HMM metrics between behavioral states significant?

Fifth, while the authors comment on having provided insights into how irreversibility relates to HMM dynamics, it is unclear what the meaning of these commonalities is, which I believe is restricted to the same or reverse pattern of irreversibility, switching rate, and max FO for the three behavioral states.

2. From text in the results (P14L5) it is unclear what the authors wish to achieve with the use of the deep autoencoder, and how the entropy production at the latent dimension of 7 is related to the irreversibility. Especially considering the HMM results, where the authors show congruent effects of irreversibility and summary metrics on only 5 states (i.e., an even lower dimension than 7).

Furthermore, they claim that “the emerging distributions match the pattern observed in the network states of the HMM and in the irreversibility at the source space”, while this only is the case for AA/REM having the highest value – QA and SWS show the reverse pattern.

3. The authors investigate the within and between region irreversibility for each region. However, neighboring electrodes are likely to have a higher correlation due to spatial leakage, and this could be affected by the configuration and number of electrodes in each region. Could the authors comment on the effect of this cross-correlation on irreversibility, and the potential confound this could be for investigating between region differences?

4. In the section “Embedded network system revealed different entropy dynamics according to the behavioral state”, the authors return to the comparison of irreversibility and the HMM that they started in the first section of the Results. This order of the results is hard to follow, and hampers readability (as mentioned in point 1). I recommend the authors to consider reorganizing the results.

5. In figure 3E-I, the authors compare the entropy production (irreversibility of HMM states) with three other information theoretic metrics, and show that these are consistent. However, it is unclear how entropy production provides more insights into the neural dynamics, and generally, what the significance is of differences in irreversibility throughout the paper.

Minor concerns

1. The authors are currently plotting differences between behavioral state in bar plots. While these are easy to read and can clearly show differences in mean between conditions, they do not display the underlying distributions of the data (see https://doi.org/10.1371/journal.pbio.1002128). Therefore, I recommend the authors use box or violin plots instead.

2. In P11L11, the authors write “Posteriorly, all 64 spectrograms were averaged into a single global spectrogram.” But later in the same section “By means of PCA the dimensionality was reduced to the 8 strongest components…”. Please clarify that (presumably) the PCA was run on the entire 64 channel data.

3. In the Methods, the definition of “< ( ) >” and “ < ( ) >” are written very concisely, but could further explained.

4. P13L24: The authors write “The frequency spectrum was used to determine the behavioral state across the whole recording”. Please add a plot of the spectrum of the three behavioral state (as a supplement), i.e., similar to Figure 3B in Stitt et al.

5. P13L21: It is unclear how the authors compute the irreversibility metric (fig. 1C) from the sensor x sensor irreversibility matrix (Fig 1B), i.e. do they average the irreversibility over channels ?

6. Figure 1D: it is unclear what the different colors correspond to. Please provide a color bar.

Reviewer #2: Thank you for inviting me to review this very interesting manuscript by Idesis and colleagues, using several markers of dynamics from thermodynamics and information theory to study ferret micro-ECoG traces across multiple behavioural states. This is a very interesting study, combining high-resolution data from a rarely-studied species, with a battery of interesting ways of characterising brain activity. The topic is of interest and timely. However, I have some reservations about the statistical reporting, which seems to be nearly absent. Discussion is instead primarily based on numerical differences. Even when statistical results are reported as non-significant, this non-significance is then simply ignored, instead repeatedly claiming that the results are consistent with previous observations when in fact the null hypothesis was not rejected. Other claims are at present difficult to evaluate because statistics are simply not provided, as far as I can determine. So my main recommendations are to add a section describing the statistical approach in detail; provide the statistics themselves in the text, and error bars in the figures; and then respect the meaning of the statistical results in the discussion and interpretation, without interpreting merely numerical differences. I believe that doing so can make for a compelling manuscript, even if the results were to be negative.

Page 3: “Furthermore, the obtained results can be compared and validated with embedded dynamics that can be obtained by dimensionality reduction approaches and generative models of switching behavior.” It would be helpful to explain why these approaches are desirable, in addition to merely stated that they can be applied.

“these data pose a unique analytical challenge – combining high 1 temporal resolution neural signals with tracking of much slower behavioral changes. Can the extracted time series metric, such as irreversibility, that is applied to the finely sampled neural recordings, reveal a precise transition between changing behaviors?”

Did the authors not already apply irreversibility to ECoG sleep data of the macaque? It would be helpful to clarify how the present study differs from that study (since both involve sleep data from non-human animals sampled at high temporal resolution). Otherwise the question appears a bit of a straw-man. The new species is a clear asset to the paper, so this may well be mentioned.

It is not clear what are the variables between which mutual information is being computed. Could the authors clarify this in the Methods?

“([F(2, 3660893)=83627,p<.01]” is the significance merely a result of an extremely high degrees of freedom? In fact, what test is this? The statistical approach does not seem to be described anywhere in the paper. It would be inappropriate to treat each instance of 1s-long recording as an independent observation, since they are in fact provided by the same animals. The statistical approach should account for this non-independence in the data, bot across conditions/physiological stages but also within each condition (same animal providing multiple observations).

“The level of irreversibility was higher in the within-region areas in all the states 29 (AA/REM: .0069; QA: .0057; SWS: .0074) compared to the between-region areas (AA/REM: 30 .0045; QA: .0052; SWS: .0063).” Is this only a numerical difference, or is it statistically reliable? I could not find statistical reporting, and Figure 2 also seems to lack error bars.

Page 20, ln. 26-30: are the associated statistics for these results provided anywhere? As far as I can tell, only numerical differences are reported, and the figure suggests that at least some of the measures are not statistically different between conditions. So as it stands, no inferences can be made based on these measures.

Page 20, “AA/REM showed the highest entropy production”. This is numerically true but statistically indistinguishable from random chance, so it is not appropriate to describe it as consistent with the irreversibility analysis: the results are in fact unable to reject the null hypothesis that behavioral state affects entropy production.

“observed a congruent pattern with previous literature comparing different levels of complexity according to the corresponding brain state (Varley et al., 2021).” I apologize if I missed this, but as far as I can determine the only measure for which any statistics are reported is the entropy production, which however was not significantly different across conditions. So in fact it is not appropriate to treat it as being consistent with the irreversibility results or with prior literature, since the difference is not distinguishable from chance, and the null hypothesis that state of wakefulness has an effect on entropy production is not rejected.

The same concern applies to lines 16-18 of page 26, where it is claimed that “we demonstrated how direct metrics derived from thermodynamics such as signal reversibility reflect brain dynamics also captured by statistical models and information theory metrics.”. The last part of this sentence is something that the authors in fact specifically failed to demonstrate, so this statement is misleading. Even if a significant difference is not found, the irreversibility results hold and are of interest in themselves.

**Have the authors made all data and (if applicable) computational code underlying the findings in their manuscript fully available?**

Reviewer #1: Yes

Reviewer #2: Yes

PLOS authors have the option to publish the peer review history of their article (what does this mean?). If published, this will include your full peer review and any attached files.

Reviewer #1: No

Reviewer #2: No
---

## [Decision Letter · Decision Letter 1]

30 Nov 2023

Dear Mr. Idesis,

Thank you very much for submitting your manuscript "Functional hierarchies in brain dynamics characterized by signal reversibility in ferret cortex" for consideration at PLOS Computational Biology.

As with all papers reviewed by the journal, your manuscript was reviewed by members of the editorial board and by several independent reviewers. In light of the reviews (below this email), we would like to invite the resubmission of a significantly-revised version that takes into account the reviewers' comments.

Thank you for submitting your revisions. Reviewer 2 reviewed the revised manuscript again, and in addition I added a third reviewer. Both reviewers find the topic of the manuscript interesting, but still need clarifications about the technical aspects of the proposed method. Moreover, they both experienced difficulties accessing the code and/or data. Therefore, I have decided that major revisions are still needed.

We cannot make any decision about publication until we have seen the revised manuscript and your response to the reviewers' comments. Your revised manuscript is also likely to be sent to reviewers for further evaluation.

Sincerely,

Fleur Zeldenrust

Academic Editor

PLOS Computational Biology

Lyle Graham

Section Editor

PLOS Computational Biology

Thank you for submitting your revisions. Reviewer 2 reviewed the revised manuscript again, and in addition I added a third reviewer. Both reviewers find the topic of the manuscript interesting, but still need clarifications about the technical aspects of the proposed method. Moreover, they both experienced difficulties accessing the code and/or data. Therefore, I have decided that major revisions are still needed.

Reviewer's Responses to Questions

**Comments to the Authors:**

Reviewer #2: I wish to thank the authors for taking my feedback into account, and for their additional work. However, I still have concerns about the appropriateness of the statistical testing and reporting.

I tried to access the data from the link provided by the authors, to have first-hand understanding of the data organization, but it seems that only a single 64x1200 matrix is shared.

I would also encourage the authors to add a dedicated Statistics section in the Methods that explains the statistical approaches in detail.

I outline my detailed feedback on the revisions below.

Response to Comment #4:

I thank the authors for the clarification of what mutual information is. However, the added text still does not explain what exactly is used as each variable, in the present case.

Response to Comment #5:

I am not convinced that a repeated-measure ANOVA is the appropriate test here. The ANOVA will (correctly) take into account that the same animals provide data for multiple conditions (AA/ Rem, SWS), but it will not take into account the fact that the same animals also provide data *within* each condition: this is an additional source of non-independence in the data that needs to be accounted for. The same issue would also apply to the new analysis with random subsamples of 1000 timepoints: the 1000 timepoints are not independent because there were not 1000 animals, so some timepoints will come from the same animal, violating the assumption of non-independence. Adding animal ID as a random effect would be one way of controlling for this violation of independence. Another way would be to average all data that come from the same animal, within each condition.

Response to Comment #6:

The figure is very helpful, thank you. However, since box-plots are used, it is not clear how many data-points were used. Throughout the text, the authors should report the statistics in full (t-score, degrees of freedom, and effect size), not just the means and p-value.

Response to Comment #7:

Bayesian statistics could be used to tell whether the difference is due to insufficient power (as the authors reasonably imply) or whether there is actual support for the null hypothesis.

Response to Comment #7:

Is the significance adjusted for all the multiple comparisons (pairwise t-tests) performed? This won’t affect the presence of overall significance from the ANOVA, but would still be appropriate when interpreting the pairwise differences. As before, I would encourage the authors to show the statistical results in full in the text or figure caption (t-score, degrees of freedom, effect size), not just the p-value, and not just for significant results.

Response to Comment #9:

As before, Bayesian statistics could be used to tell whether the difference is due to insufficient power or whether there is actual support for the null hypothesis.

Minor

Figure in response to Comment #6: there is the word “path” in violet, which the authors may wish to remove.

Reviewer #3: Dear authors, Please find attached my review

**Have the authors made all data and (if applicable) computational code underlying the findings in their manuscript fully available?**

Reviewer #2: **No: **I tried to access the data from the link provided by the authors, to have first-hand understanding of the data organization, but it seems that only a single 64x1200 matrix is shared in the Data directory.

Reviewer #3: **No: **The data and the codes are not available.

PLOS authors have the option to publish the peer review history of their article (what does this mean?). If published, this will include your full peer review and any attached files.

Reviewer #2: No

Reviewer #3: No
---

## [Decision Letter · Decision Letter 2]

9 Jan 2024

Dear Mr. Idesis,

We are pleased to inform you that your manuscript 'Functional hierarchies in brain dynamics characterized by signal reversibility in ferret cortex' has been provisionally accepted for publication in PLOS Computational Biology.

Best regards,

Fleur Zeldenrust

Academic Editor

PLOS Computational Biology

Lyle Graham

Section Editor

PLOS Computational Biology

Both reviewers agree that the manuscript is much clearer now, and ready for publication.

Reviewer's Responses to Questions

**Comments to the Authors:**

Reviewer #2: The authors have satisfactorily addressed all my comments.

Reviewer #3: The revised manuscript exhibits a remarkable enhancement in clarity, rendering it more accessible, as formulas and specific phrases have been corrected. Notably, the improvements made to the figures have significantly strengthened the main message, contributing to a more cohesive narrative and providing a clearer visual representation of your findings.

**Have the authors made all data and (if applicable) computational code underlying the findings in their manuscript fully available?**

Reviewer #2: Yes

Reviewer #3: Yes

PLOS authors have the option to publish the peer review history of their article (what does this mean?). If published, this will include your full peer review and any attached files.

Reviewer #2: No

Reviewer #3: No

---

## [Editor Report · Acceptance letter]

17 Jan 2024

PCOMPBIOL-D-23-01038R2 

Functional hierarchies in brain dynamics characterized by signal reversibility in ferret cortex

Dear Dr Idesis,

I am pleased to inform you that your manuscript has been formally accepted for publication in PLOS Computational Biology. Your manuscript is now with our production department and you will be notified of the publication date in due course.

With kind regards,

Anita Estes
